# Asthma and Obesity: Two Diseases on the Rise and Bridged by Inflammation

**DOI:** 10.3390/jcm10020169

**Published:** 2021-01-06

**Authors:** Marina Bantulà, Jordi Roca-Ferrer, Ebymar Arismendi, César Picado

**Affiliations:** 1Department of Internal Medicine, Hospital Clinic, Institut d’Investigació Biomèdica August Pi i Sunyer (IDIBAPS), 08036 Barcelona, Spain; bantula@clinic.cat (M.B.); jrocaf@clinic.cat (J.R.-F.); earismen@clinic.cat (E.A.); 2Department of Medicine, University of Barcelona, 08036 Barcelona, Spain; 3Centro de Investigaciones Biomédicas en Red de Enfermedades Respiratorias (CIBERES), 08036 Barcelona, Spain; 4Servei de Pneumologia, Hospital Clinic, 08036 Barcelona, Spain

**Keywords:** asthma, cytokines, phenotype, inflammation, obesity

## Abstract

Asthma and obesity are two epidemics affecting the developed world. The relationship between obesity and both asthma and severe asthma appears to be weight-dependent, causal, partly genetic, and probably bidirectional. There are two distinct phenotypes: 1. Allergic asthma in children with obesity, which worsens a pre-existing asthma, and 2. An often non allergic, late-onset asthma developing as a consequence of obesity. In obesity, infiltration of adipose tissue by macrophages M1, together with an increased expression of multiple mediators that amplify and propagate inflammation, is considered as the culprit of obesity-related inflammation. Adipose tissue is an important source of adipokines, such as pro-inflammatory leptin, produced in excess in obesity, and adiponectin with anti-inflammatory effects with reduced synthesis. The inflammatory process also involves the synthesis of pro-inflammatory cytokines such as IL-1β, IL-6, TNFα, and TGFβ, which also contribute to asthma pathogenesis. In contrast, asthma pro-inflammatory cytokines such as IL-4, IL-5, IL-13, and IL-33 contribute to maintain the lean state. The resulting regulatory effects of the immunomodulatory pathways underlying both diseases have been hypothesized to be one of the mechanisms by which obesity increases asthma risk and severity. Reduction of weight by diet, exercise, or bariatric surgery reduces inflammatory activity and improves asthma and lung function.

## 1. Introduction

Asthma is a common chronic airway disease characterized by variable airflow limitation resulting from the combination of airway narrowing, airway hypersensitivity, airway wall thickening, and increased mucus hypersecretion. Airway narrowing results from both chronic inflammation and airway remodeling. Asthma is a heterogeneous disease with several distinct clinical presentations (phenotypes) and complex pathophysiological mechanisms (endotypes) [1].

Obesity is the consequence of an excessive body fat accumulation due to an imbalance of energy intake and energy expenditure [2]. Diagnosis of obesity is usually established by assessing the body mass index (BMI) ≥ 30 kg/m^2^. However, it is likely that the BMI measured by the formula: Mass (in kilograms) divided by size (in meters) squared is not the best marker of obesity. This anthropometric measurement has some limitations, because the correlation between body fat and BMI is not constant. Assessing body fat distribution with measurements such as waist circumference or waist-to-hip ratio may improve evaluation and diagnosis of obesity [2]. Currently, DualEnergy X ray Absorptiometry-DEXA, is considered as the reference method in the assessment of body composition [3]. The method used to characterize asthma and obesity may have a relevant impact on the results of studies aimed at evaluating the association of asthma and obesity. A recent study showed that the prevalence of obesity in children can vary widely, depending on the criteria used to assess body mass. These differences may explain the discrepancies reported in clinical and epidemiological studies regarding the strength of the evidence supporting the association of asthma and obesity (see below) [4].

Recent studies suggest that the interaction between obesity and asthma is more complex than has been reported so far and goes beyond a simple association between excess weight and asthma. In this line, there are studies that show that the characteristics of the diet, such as the acid load of the diet [5] or the exposure to some chemical compounds and indoor and outdoor pollutants can also contribute to the association of obesity and asthma [6,7,8].

The objective of this review is to provide an update on the epidemiological studies that support the association of asthma and obesity and to describe the potential role of the inflammatory mechanisms underlying this association. We will also discuss the effects of weight loss on the inflammatory mechanisms of the asthma-obesity phenotype.

## 2. Asthma and Obesity Two Diseases on the Rise and Linked

Numerous epidemiological studies have reported the significant increase of asthma and obesity in most countries all over the world.

### 2.1. Asthma

Asthma is global health problem affecting almost 300 million people of all ages and ethnic groups worldwide [9]. In Spain, the prevalence of asthma ranges between 1.5 and 16.7% in the adult population, and about 10% in the pediatric population [10].

The prevalence of asthma has markedly increased over the last decades, especially in Westernized countries [11,12]. The causes of this increase are unclear but may be a consequence of changes in lifestyle and in environmental conditions. Lifestyle changes include modification of dietary habits, with decreased consumption of vegetables and fresh fruits, and increased access to processed saturated fats and carbohydrate sweeteners [13,14,15]. Several mechanisms have been proposed to explain the role of diet in asthma, including low vitamin D levels, oxidative stress, epigenetic regulation, and imbalance in the gut microbiome [13]. The environmental changes contributing to the increase in the prevalence of asthma include increased exposure to tobacco smoke, traffic pollution, and infectious agents [16]. However, the underlying mechanisms involved in the interaction of environmental pollution, allergen, and viral exposures with the immune system remains to be elucidated.

### 2.2. Obesity

Worldwide, the prevalence rate for being overweight or obese between 1980 and 2013 increased 27.5% for adults and 47.1% for children, resulting in a total of 2.1 billion individuals considered overweight or obese [17]. In Spain, the age-adjusted prevalence for being overweight or obese increased from 34.0% to 35.8%, and from 8.0% to 16.5%, respectively, between 1987 and 2012. Morbid obesity increased from 0.20% in 1993 to 0.88% in the same period. The growth rate was greater among males [18].

The increase in the prevalence of obesity has been associated with factors favoring a positive energy balance and weight gain, which include increased food consumption, particularly of high-calorie foods, and decreased time spent in occupational physical activities associated with increased sedentary activities. Various studies have shown that the rate of heritability of BMI ranges from 40 to 70%, supporting the existence of an underlying genetic mechanism that contributes to obesity. Over 300 loci have been identified, although these loci only account for less than 5% of individual variation in BMI [19,20].

### 2.3. Asthma and Obesity: Two Linked Diseases

The epidemiological link between asthma and obesity was first suggested in a study carried out by Camargo et al., involving 85,911 nurses in the United States [21]. The study found that the risk of developing late-onset asthma was significantly increased when the BMI was greater than or equal to 30 kg/m^2^ with an odds ratio of 2.6. Subsequently, several studies were carried out and corroborated the existence of an excess risk of developing asthma in obese subjects compared with subjects not overweight, regardless of gender or age [22,23,24,25,26,27,28,29]. The relationship also appears stronger for those with central versus general adiposity [30,31]. A study from the California Teachers Study cohort reported that increased waist circumference was associated with asthma even among those with BMI’s within the normal range [30]. The European Community Respiratory Health Survey (ECRHS) found an association between asthma and obesity, but with a greater excess risk in females with respect to males [23,32]. The longitudinal cohort conducted in the city of Tucson in the United States [33] reported the persistence of symptoms into adulthood in obese children with asthma, and also found that being overweight is an independent risk factor for developing post-puberty asthma. Research has also shown associations between mothers’ overweight status just before and during pregnancy and offspring’s asthma [34,35]. A recent study found that fathers who were overweight during adolescence predispose their offspring to develop asthma [36].

Interestingly, a recent study suggests that asthma may also lead to obesity. The authors followed non-obese children for up to 10 years. Children with an initial diagnosis of asthma were approximately 50% more likely to become obese than children without asthma. The study also showed that the increased risk of obesity with asthma was driven by children who were already overweight at baseline. Since being overweight is a very strong predictor of subsequent obesity, these observations support the possibility that being overweight may drive both asthma and subsequent obesity, and that both obesity and asthma can interact by complex and multiple mechanisms [37].

### 2.4. Severe Asthma Is Associated with Obesity

Various recent studies that analyzed the profile of mild to severe asthma patients highlighted a subgroup of obese asthma patients, more often female, with late-onset, non-atopic asthma. Asthma in these patients is more difficult to control, lung function is impaired, they undergo more treatments with limited therapeutic effects, and suffer from more frequent exacerbations than the non-obese asthma population [38,39,40,41,42,43]. The frequent use of systemic corticosteroids that characterizes severe asthma may contribute to the development of obesity in these patients [44].

Two main sub-phenotypes of obese asthma have been described according to age. The first phenotype associating asthma and obesity of early onset affects children with asthma under 12 years old. It is characterized by obesity, which worsens pre-existing asthma [45]. Children are, in the majority of cases, allergic, and both sexes are affected equally. Asthma is associated with more severe airway obstruction, more marked airway hyperresponsiveness (AHR) compared with early non-obese asthma, and inflammation is predominantly eosinophilic [46]. The second main phenotype is characterized by delayed onset asthma developing, at least in part, as a consequence of obesity [44]. This asthma is generally non-allergic with pronounced symptomatology despite treatments with high doses of inhaled corticosteroids and long-acting bronchodilators [40,41]. There are, however, some discrepancies with respect to the type of airway inflammation associated with the obese phenotype. Some cluster analyses have shown that the association between obesity and neutrophilic inflammation is more common in women [41]. However, other studies have not found such an association between obesity and neutrophilic airway inflammation in adults with asthma [47,48].

## 3. Asthma and Obesity Two Diseases Bridged by Inflammation

### 3.1. Asthma and Inflammation

Asthma can be classified as allergic and non-allergic, eosinophilic, and non-eosinophilic, and type 2 (T2) high and T2 low or its equivalent non-T2, regarding the inflammatory profile [38,40,41,49]. The observation that, besides T helper type 2 (Th2) cells, other innate immune cells like type 2 innate lymphoid cells (ILC2) can produce Th2-cell-associated cytokines in asthma, has motivated the gradual shift in terminology from ‘‘Th2 asthma” to ‘‘T2 asthma” [49]. Given that the majority of studies on the obese asthma phenotype use the eosinophilic, non-eosinophilic terminology, in this review, we will use this classification.

#### 3.1.1. Eosinophilic Asthma

The diagnosis of eosinophilic asthma is currently based on the demonstration of elevated numbers of eosinophils in induced sputum. Surrogate biomarkers of eosinophilic asthma include blood eosinophils and fractioned exhaled nitric oxide (FeNO), when used in combination they provide the highest diagnostic accuracy for ruling in or out eosinophilic asthma [50].

Recent studies have demonstrated the importance of both innate and adaptive immunity in the immunologic mechanisms in asthma [51]. T2 asthma with eosinophilia is a common phenotype in asthma. It occurs with and without any demonstrated allergy.

The type 2 endotype with allergy is the most common asthma phenotype with an early onset. Type 2 immune responses involves Th2 cells, ILC2, immunoglobulin E (IgE)-producing B secreting cells, natural killer T (NK-T) cells, mast cells, basophils, eosinophils, and their cytokines. Th2 cells produce various cytokines such as interleukin (IL)-4 (IL-4), IL-5, IL-9, and IL-13 [51,52,53,54,55]. Whereas IL-4 is important for the allergen-specific synthesis of IgE, IL-5 is crucial for the recruitment and survival of eosinophils [51,54,56]. IL-9 is made by a subset of CD4+ T cells (Th9 cells), impairs IFN-γ production, and promotes IL-4-induced IgE secretions, and its serum levels have been found to be high in allergic asthma [55]. ILC2 produce type 2 cytokines such as IL-4, IL-5, IL-9, and IL-13, and together with other immune cells expressing the receptor suppression of tumorigenicity 2 receptor (ST2) for IL-33 are involved in the development of eosinophilic allergic asthma [57,58,59].

Eosinophilic non-allergic asthma is in most cases a moderate to severe, late-onset disease frequently associated with chronic rhinosinusitis and nasal polyps. Eosinophil non-allergic asthma is an ILC2 predominant process. Cytokine production from ILC2 is controlled by IL-10, transforming growth factor beta (TGF-β) and alarmins, such as IL-25, IL-33, and thymic stromal lymphopoietin (TSLP). Alarmins act as intercellular signals by interacting with chemotactic and pattern recognition receptors (PRRs) to boost immune cells in host defese [51]. Moreover, they have the ability to activate dendritic cells (DC) to maturity and to interact with adaptive immunity and T cell-dependent long-term immune memory [58,59]. Alarmins are released mainly by airway epithelial cells when triggered by the exposure to toxic agents, pollutants, diesel particles, tobacco smoke and virus, they contribute to the inflammatory airway response in eosinophilic non-allergic asthma. Release of IL-25 and IL-33 by damaged epithelial cells activates ILC2 to produce Th2-cytokines [51,58,60,61]. TSLP promotes chemotaxis and activation of eosinophils by delaying their apoptosis and enhancing the activity of the adhesion molecule machinery involved in eosinophil recruitment and chemotaxis [62]. Combined stimulation with TSLP and IL-33 elicited an approximate 10-fold increase in cytokine production by ILC2 compared with stimulation by IL-33 alone [63,64]. Recent studies have demonstrated that IL-33 levels in sputum, serum, and tissue expression correlated with asthma severity [65,66]. IL-33 together with TSLP and IL-25 provide a signal for innate ILC2, which in turn initiate and amplify allergic inflammation by orchestrating the T2 immune response [67]. TSLP has been found to be overexpressed in the airways of human patients with severe asthma [68,69], and has been linked to steroid-resistant asthma [70] When blood ILC2 are exposed to TSLP, they become steroid resistant. Moreover, dexamethasone cannot suppress cytokine production by ILC2, obtained from patients with severe asthma. It is currently estimated that only about half of asthma patients have evidence of T2 immunity in their airways [49].

#### 3.1.2. Non-Eosinophilic Asthma

Non-eosinophilic asthma has been described mostly in adults and rarely in children and comprises neutrophilic and paucigranulocytic asthma [71,72].

Th1 and Th17 are the subset of cells that produce IL-17, IL-21, and IL-22, the dominant cytokines in neutrophilic asthma [73,74]. IL-17 is responsible for the recruitment of neutrophils into the lungs [75]. Activation of the neutrophils takes place by the production of IL-6, granulocyte colony-stimulating factor (G-CSF), granulocyte-macrophage colony-stimulating factor (GM-CSF), IL-8, chemokine (C-X-C motif) ligand 1 (CXCL1), CXCL5, and CXCL8 from airway epithelial cells [73,74]. IL-17 is associated with neutrophilic inflammation, AHR, and severe asthma with corticosteroid resistance [76,77,78,79,80]. IL-23 is the cytokine responsible for maintaining the Th17 cell functionally active [81]. Type-3 innate lymphoid cells (ILC3), other IL-17 secreting ILC, and pro-inflammatory macrophages also appear to play a key role in neutrophilic corticosteroid-resistant asthma [82].

Another mechanism that also accounts for neutrophilic airway inflammation in asthma is inflammasome activation. The nucleotide-binding oligomerization domain-like receptor family, pyrin domain containing 3 (NLRP3) inflammasome, is an intracellular multiprotein complex that facilitates the autoactivation of pro-inflammatory cysteine protease caspase-1. Then, the activated caspase-1 cleaves pro-IL-1β and pro-IL-18 into their mature forms. Active IL-1β was found to promote Th17 cell dependent inflammation [83]. NLRP3 is activated by danger-associated molecular patterns (DAMP) and serum amyloid A protein which is produced by epithelial cells exposed to microbes [83]. Gene expression of NLRP3, IL-1β, and caspase-1 are detected at high levels in sputum and peripheral blood of asthma patients with neutrphilic airway inflammation [84]. In addition, IL-1β levels correlate with sputum IL-8 levels in patients with neutrophilic asthma [85].

Th1 related cytokines such as IFN-γ and tumor necrosis factor alpha (TNF-α) have been found increased in patients with severe neutrophilic asthma [86,87]. High IFN-γ levels in the airways promote AHR via the suppression of secretory leukocyte protease (SLPI) [86]. The amount of TNF-α is also increased in the airways of patients with severe steroid-resistant asthma [88]. However, in clinical trials, the results of TNF-α blockade have been variable and are questionable [87,89,90].

IL-6 is a pro-inflammatory cytokine excreted from many different cell types including T cells, myeloid lineage cells, and endothelial cells [91]. In contrast, the IL-6 membrane-bound receptor (IL-6R) has a limited expression distribution in immune cells, including T cells and neutrophils [91]. IL-6 can bind either IL-6R (classical pathway) or glycoprotein 130 after forming a complex with the soluble IL-6 receptor (sIL-6R) (trans-signalling pathway). Neutrophils are the main source of IL-6 generated in the airways of asthmatic patients [92]. Increased levels of IL-6 have been found in sputum, serum, and bronchoalveolar fluid (BALF) of asthmatic patients. IL-6 has been found associated with severe asthma in adults [93,94], but not in children [95]. Patients with asthma who have high circulating concentrations of IL-6 have a much more severe asthma than those without predominant IL-6 inflammation [96]. It has also been reported that IL-6 levels in sputum are inversely correlated with the predictive percentage of Forced Expiratory Volume 1 second (FEV_1_) [96]. High IL-6R mRNA and IL-6 protein sputum levels have been found associated with higher sputum neutrophils in patients with severe asthma [97]. This group also has poor lung function and higher levels of systemic IL-6 [97]. A recent study has described a novel asthmatic patient subset with IL-6 trans-signalling (IL-6TS) pathway activation in the lung epithelium in the absence of systemic IL-6 inflammation. These patients are characterized by an increased exacerbation rate, T2 inflammation–independent eosinophilia, increased markers of submucosal inflammation, airway remodeling, and decreased expression of epithelial junction components [94].

### 3.2. Obesity and Inflammation

Chronic obesity induces low-grade inflammation in AT called “metainflammation”, which is mainly mediated by macrophages without being related to infection or tissue damage [98,99].

#### 3.2.1. Adipose Tissue Expansion and Inflammation

The mechanisms triggering AT metainflammation have yet to be elucidated. It is generally accepted, however, that adipocyte hypertrophy and hyperplasia occurring in obese individuals may play an important role [99]. As AT expands and the distance between adipocytes and capillaries increases, hypoxic death of some adipocytes occurs. There are two types of macrophages: M1 and M2. M1 is known to stimulate pro-inflammatory factors and induce insulin resistance. In contrast, M2 is known to block an inflammatory response and promote oxidative metabolism. In response to adipocyte death, pro-inflammatory macrophages M1 surround dead and dying cells and remove debris from the damaged area. During this process, adipocytes and M1 produce inflammatory cytokines including IL-6, TNF-α, IL-1β, and monocyte chemoattractant protein (MCP-1). The number of AT macrophages in humans is low (4%) and increases to 12% when developing excess adiposity [100]. Infiltration of AT by M1 together with their altered function and anatomical localization is nowadays considered the culprit of obesity-related metainflammation. Cytokine production does not resolve the problem which becomes chronic, and leads to impaired adipocyte insulin signaling, further inflammation, and a continued worsening of AT dysfunction [99].

Serum IL-6 levels have been found to be higher in obese subjects and correlated with all indexes of obesity [101,102] and with visceral adipocytes [103].

#### 3.2.2. Innate and Adaptive Immune Systems

A variety of cell types from both the innate and adaptive immune systems have been found in AT playing important roles in tissue homeostasis maintenance under non-obese conditions [104]. Myeloid cells, considered the main players in innate immunity and as macrophages, are the most abundant immune cell type in AT and their infiltration forms the basis of AT inflammation. However, excessive fat accumulation leads to substantial changes in the amount and function of other immune cells increasing the number and activity of some of them (mast cells, neutrophils, and T- and B lymphocytes) while simultaneously reducing others, including eosinophils and several subsets of T lymphocytes (T helper 2 (Th2), regulatory T cells (Treg), and invariant natural killer T cells (iNKT)) [105,106]. As lymphocytes are the second-largest immune cell fraction in obese AT with changes in amount and activity occurring even before those in macrophages, it seems that adaptive immunity also takes its turn in the processes of metainflammation [105,106].

Interestingly, the four cytokines with a prominent pro-inflammatory role in the asthmatic lung -IL-4, IL-5, IL-13, and IL-33- contribute to maintain the lean state. IL-4 and IL-13 are produced by ILC2 cells and eosinophils and trigger M2 macrophages to express two anti-inflammatory cytokines, TGF-β and IL-10. TGF-β inhibits adipogenesis, and IL-10 maintains insulin sensitivity in adipocytes [107,108]. IL-33 maintains ILC2 cells and reduces the risk of metabolic syndrome [109]. In humans, low serum IL-33 levels are associated with high body mass index [110]. IL-33 is able to counter excessive inflammation in AT responses by targeting immune cells expressing the ST2 receptor. Two isoforms of ST2 have been identified, the full-length receptor (ST2l) and the soluble ST2 (sTS2). The sST2 functions as a decoy receptor capable of abrogating IL-33 signaling. Mice lacking ST2 or IL-33 develop increased adiposity and worsened metabolic profiles. IL-33 treatment triggered the expansion of a group of Fox3+ST2+T_regs_ and attenuated AT inflammation [110].

While respiratory studies associate eosinophils and ILC2 cells with inflammation in asthma, these cells maintain the tolerant adipose microenvironment [111,112].

Eosinophils are normally present alongside adipocytes and other resident adipose leukocytes and are the major source of IL-4 in adipose tissue. It has also been suggested that eosinophils may mediate glucose homeostasis and energy expenditure. However, the role of eosinophils in obesity is a matter of debate with studies showing conflicting findings, some suggesting that they protect from obesity while others suggest the opposite [113].

In the AT of obese individuals, plasma concentrations of myeloperoxidase (MPO) and calprotectin (a factor mainly derived from neutrophils) as well as the levels of neutrophil activation marker CD66b were found to be increased compared with lean controls, suggesting that obesity also affects systemic activation of neutrophils [114].

#### 3.2.3. Adipokines

Visceral AT is an important source of cytokine production, also called adipokines, and is a crucial factor in metainflammation in obese individuals [102]. Some adipokines such as leptin or resistin are produced in excess in obesity while others, such as adiponectin, are reduced [115].

Adiponectin has three different molecular weight isoforms: low-molecular weight (LMW), middle (MMW), and high (HMW) [116,117]. MMW and HMW constitute the majority of circulating adiponectin while LMW is present at very low concentrations in human plasma [116,117]. Adiponectin plays its physiological role via the activation of AdipoR_1_ and AdipoR_2_ receptors [118]. Early studies have shown that adiponectin possesses anti-inflammatory properties [115,116]. However, results from recent investigation supports that adiponectin can also exert pro-inflammatory actions in some diseases such as rheumatoid arthritis (RA), chronic kidney disease, inflammatory bowel disease, and autoimmune diseases [116,117]. On macrophages, adiponectin promotes cellular differentiation of monocytes to M2 macrophages and suppresses their differentiation to M1 macrophages [116,117]. Adiponectin activates anti-inflammatory IL-10 and reduces pro-inflammatory cytokines such as IFN-γ, IL-6, and TNF-α in human macrophages [119]. Circulating serum levels of adiponectin are decreased in patients with obesity, type 2 diabetes, metabolic syndrome, or cardiovascular disease inflammation, and are associated with an increased release of pro-inflammatory cytokines like IL-6 and TNF-α [120]. It was recently demonstrated that adiponectin has a protective role in the murine inflammatory response, leading to decreased neutrophil recruitment and decreased expression of cytokines and chemokines, especially IL-17 [121]. When weight is lost, the adiponectin serum level increases, and this has a positive relationship with BMI reduction [122].

In contrast to the anti-inflammatory effects, high serum levels of adiponectin results in systemic chronic inflammation in RA. Adiponectin stimulates the production of pro-inflammatory factors such as IL-6 and IL-8 in synoviocytes of RA patients [123,124]. Similarly, various studies have reported that the total adiponectin level in patients with end stage renal disease is higher than that of the control group [125]. Furthermore, chronic kidney disease patients have systemic low-grade chronic inflammation and adiponectin seems to play a key role in triggering renal injury [126]. In addition, other studies showed that patients with systemic autoimmune diseases have elevated adiponectin serum levels [127].

The reason that adiponectin may have pro-or-anti-inflammatory effects remains to be elucidated. Some observations support that the opposing roles of adiponectin is in part determined by the predominant molecular isoform involved in adiponectin activity. Results of some studies indicate that adiponectin isoforms have differential effects on inflammation: LMW has anti-inflammatory effects, while HMW seems to activate pro-inflammatory factors [128,129,130].

Leptin is an adipokine mainly produced by adipocytes with a dual role as a hormone and as a cytokine. As a hormone, it has a key function in the regulation of food intake and energy expenditure and as a cytokine exerts strong pro-inflammatory activities [131]. In humans, four splice variants of the leptin receptor have been identified: A long isoform and three short isoforms [132,133]. The long isoform is responsible for the anorexigenic effects of leptin, it is abundant in the hypothalamic centers regulating food intake and can also be found on immune cells [134,135].

Obesity is characterized by elevated leptin levels, as well as by resistance to the anorectic effects of leptin [136,137]. Early leptin research showed an over-expression of the leptin gene in AT and a strong positive association between serum leptin concentrations and the percentage of body fat in obese individuals. Several mechanisms and pathways accounting for the development of leptin resistance have been described [136,137].

As a cytokine, leptin stimulates adipocytes to secrete pro-inflammatory mediators such as TNF-α, IL-6, [138,139], MCP-1 [140,141,142] and IL-12 [61]. An in vitro study showed that, in a concentration-dependent manner, leptin could activate human peripheral blood mononuclear cells (PBMC) to induce secretion of IL-6 and TNF-α [143]. Higher levels of TNF-α and IFN-γ have been found in PBMC culture from individuals with a BMI ≥ 30 kg/m^2^ compared with those with BMI of less than 30kg/m^2^, and this was associated with an increased leptin blood concentration in these obese subjects [144]. Leptin promotes naive T cell survival and facilitates the differentiation and activity of Th1 cells while inhibiting the cytokine production of Th2 cells (IL-4, IL-5, and IL-10) [142,145,146,147], enhances the proliferation and activation of T cells, and exerts differential effects on the proliferation of naïve versus memory T cells or effector T cells versus Tregs [148,149]. It also promotes the activity of pro-inflammatory Th17 cells [150].

#### 3.2.4. Inflammasome

One possible mechanism that translates obesity into chronic inflammation has been recently identified in the inflammasomes. NLRP3 is activated by saturated fatty acids such as palmitate and stearate, as well as free cholesterol and cholesterol crystals, and by oxidative stress, which is also known to be present in AT in obesity [151]. During caloric excess, NLRP3 activation results in caspase-1 activation, which can in turn cleave pro-IL-1β, and M1-type macrophages are able to secrete pro-inflammatory cytokines such as IL-1β, IL-18, MCP-1, TNF-α, and IL-6 into the circulation and act together with other secreted inflammatory adipokines, driving inflammation to many organs [61].

### 3.3. Inflammatory Links between Asthma and Obesity

Asthma and obesity share some of the mechanisms responsible for their underlying inflammatory process. This finding raises the possibility that additive or synergistic effects between both inflammatory processes may account, for instance, for the reported association of severe asthma with obesity. The increased production of some pro-inflammatory cytokines by the adipose tissue in obese asthmatics could have clinical and lung function consequences for these patients. Various observations appear to support this possibility.

#### 3.3.1. Cells and Cytokines

Several studies have evaluated the cell dominant pattern of airway inflammation in asthma patients with and without associated obesity. Data from these studies support that obesity is associated with a neutrophil dominant rather than eosinophil dominant inflammatory pattern in the airway lumen [37,152,153,154]. The abundant neutrophil concentration is associated with the presence of greater levels of IL-17A, a cytokine involved in neutrophil recruitment to the airways [37,152,153]. Data from animal models also support a link between IL-17A and obese asthma. Obese mice typically exhibit innate AHR, but this AHR is not observed when the animals are IL-17A deficient [155]. In the lungs of obese mice, increased IL-17A, producing CCR6+ ILC3, was found associated with AHR and neutrophilic inflammation [155]. Obese mice with a deletion of the TNF-α receptor (TNFR2) were protected against innate AHR and presented reduced levels of IL-17 in comparison with controls [156].

Interestingly, the relative reduction of eosinophil numbers with respect to neutrophils in the airway secretions of obese asthmatics contrasts with the higher eosinophil counts found in the airway submucosa in obese versus lean severe asthmatics [152,157]. Similar to IL-17, sputum IL-5 and IL-25 levels have been found to be significantly higher in obese asthmatics compared with their lean counterparts. In contrast, neither IL-4 nor IL-13 sputum levels were found associated with BMI in asthma patients [152]. Two hypotheses have been suggested to explain the apparent paradox represented by the reduced presence of eosinophils in sputum versus elevated eosinophilia in submucosa: 1. Survival of eosinophils in the airway is reduced in obese asthmatics and 2. Eosinophils are retained in the submucosa and do not migrate to the airway lumen. The second hypothesis appears to be supported by studies reporting that more eosinophils are recruited to the lungs of obese patients with asthma compared with non-obese patients with asthma [158]. Furthermore, it is unlikely that eosinophils fail to survive within the airway lumens of obese asthmatics because IL-5, a well-known eosinophil survival factor, is elevated in the sputum of obese versus lean severe asthmatics [152,157]. This observation is interesting because some severe obese asthmatics may have an eosinophilic (in airway submucosa) non-T2 dominant type of asthma, and therefore may benefit from eosinophil targeted therapeutics that might be excluded if attending only sputum results.

The obesity-related asthma phenotype is also associated with the presence of increased interleukins levels, such as TNF-α and IL-1β in the lung, even in the absence of an antigenic challenge [159]. TNF-α expression increased in PBMC in parallel with BMI increase in subjects with asthma [46].

Given the relevant role of IL-6 in obesity and some asthma endotypes, numerous studies have examined the potential bridging role of IL-6 for these two conditions and have reported contradictory findings. Serum IL-6 level significantly increased with BMI percentile in children, but no relationship was found with asthma severity [95]. However, a significant association was observed between baseline IL-6 level and the probability of experiencing an asthma exacerbation treated with systemic corticosteroids during the 1-year study. The odds of experiencing at least one exacerbation during the study increased by 24% for each quartile increase in serum IL-6. The same study could not find any association between IL-6 levels with markers of T2 inflammation, including total blood eosinophils, total IgE, or number of allergen sensitizations [95].

Similar results were described in peripheral blood measurements, where neutrophil counts and IL-6 were significantly increased in the morbid obese adult asthma group compared with the non-obese group [160]. In asthma, elevated serum IL-6 has been found associated with increased body weight, with lower lung function and greater exacerbation risk independent of obesity [93] (Figure 1).

#### 3.3.2. Inflammasome Role in Asthma-Obesity Phenotype

IL-1β is found elevated in the blood of obese individuals [161]. The release of IL-1β from cells is dependent upon the activation of caspase-1 and its assembly with the NLRP3 inflammasome, which can be activated by fatty acids via toll-like receptor 4 (TLR4). Increased sputum concentrations of IL-1β and increased NLRP3 and TLR4 expression in sputum cells has been reported in obese versus non obese asthmatic patients [162,163].

A recent paper reported that obesity induced by a high-fat diet in mice triggered the activation of an NLRP3 inflammasome in M1 macrophages resident on adipose tissue and in the lungs, resulting in an amplification of IL-1β production, the subsequent ILC3 activation, and IL-17 secretion, which in turn facilitates AHR in these patients [61]; this is a novel mechanism that has not been previously linked with airway disease [159].

#### 3.3.3. Adipokines

Low concentration of serum adiponectin is associated with higher asthma incidence according to a recent meta-analysis of 13 studies [127]. The meta-analysis found that, in the overall study population, the diagnosis of asthma was associated with lower levels of adiponectin in patients with asthma compared with controls. However, borderline association of adiponectin with asthma was seen in adults, but not in children. The study also shows that higher leptin levels were associated with asthma both in adults and children.

Recent findings suggest that the leptin pathway may partly explain the obesity asthma relationship. Sideleva et al. found that increased leptin levels are associated with AHR [164]. Leptin treatment augmented allergen-induced AHR but did not affect eosinophil influx or Th2 cytokine expression, suggesting that leptin is capable of augmenting AHR through a mechanism independent of Th2 inflammation. Rather than modifying adaptive immunity, leptin could be acting on the innate immune system: Exogenous administration of leptin to lean mice increases their subsequent inflammatory response to acute O_3_ exposure [165], a response characterized by the release of acute-phase cytokines and chemokines, and dependent to some extent on TLR activation [166].

A few cross-sectional studies have reported positive associations of serum leptin concentration with asthma severity, asthma control, lung function, and asthma severity in children and in adults [167,168]. However, the cross-sectional design does not allow for the establishment of whether the modification in the biological marker concentration is a cause or a consequence of the disease. Two recent studies examined the potential role of leptin in obese asthmatics using analytical methods, which help elucidate the possible causal role of leptin in the association between obesity and asthma. One study found an association between serum leptin levels and asthma control assessed by the Asthma Control Questionnaire (ACQ) [169]. The second recent longitudinal study reported an indirect effect mediated by leptin in the association between adiposity and persistent asthma [170]. Collectively, these results support that leptin may be a mediator that contributes to explain the association between obesity and both asthma persistence and control (Figure 1).

## 4. Association of Obesity and Asthma: Genetics

A genetic predisposition has been suggested to explain why some obese subjects will develop asthma while others will not. A study on mice and human subjects found that CHI3L1 gene expression and the protein generated by its activation (chitinase 3-like 1) can be induced by a high-fat diet and thereby contribute both to obesity and to asthma development [171].

A recent study performed a large genome-wide association study (*n* > 450,000) to explore the genetic associations between obesity and early- versus late-onset asthma in an adult population, and between obesity and atopic versus non-atopic asthma. Limited evidence of shared genetic correlation between BMI and early-onset asthma was found. However, the results of the study were able to confirm causal effects of BMI on late-onset, atopic, and non-atopic asthma and identified 32 independent shared loci between these traits and the HLA (human leukocyte antigen) region, ERBB3 (regulation of bronchial epithelial repair and remodelling), and SMAD3 (regulation of inflammatory response) genes. These results provide support to the existence of a causal link between obesity and asthma. Moreover, the shared loci identified support the involvement of inflammation, airway repair, and the immune system in the underlying pathophysiological mechanisms shared by obesity and asthma [172].

## 5. Obesity, Metabolic Syndrome, and Asthma

Most of the studies designed to examine the interactions between asthma and obesity were based on select cohorts of obese children and adults. However, obesity contributes to the development of metabolic syndrome, a constellation of health risk factors that includes dyslipidemia, insulin resistance, type 2 diabetes, hypertension, and expression of pro-inflammatory mediators. There is new interest in the relationship of asthma with other inflammatory mechanisms related to the metabolic syndrome. It is well-known that only a subset of obese individuals with central adiposity develops insulin resistance, type 2 diabetes, and systemic inflammation. Some investigations support that it is this subset of obese individuals that are at risk for asthma, and that components of the metabolic syndrome or insulin resistance underlie the pathogenesis of asthma [173,174]. Hyperinsulinemia may lead to changes in the lung characteristics of asthma and appears to be an independent risk factor for asthma in some studies [175]. Interestingly, a recent study found that asthma was directly associated with elevated serum triglyceride levels and insulin resistance regardless of BMI. This observation is relevant because children are at risk of being overlooked because they present a healthy appearance based on weight and adiposity, however, their metabolism is already abnormal and predisposes them to asthma similarly to their overweight peers [176].

A recent study examined the relationship between IL-6 plasma levels with asthma and obesity. Plasma IL-6 was significantly and positively correlated with BMI. However, 62% of obese patients had normal plasma IL-6, an observation that prompted the authors to examine the relationship between IL-6 inflammation, metabolic dysfunction, and asthma severity in obese and non-obese patients. With this objective they compared outcomes of metabolic dysfunction and asthma severity in IL-6-high and IL-6-low asthma, and stratified patients into obese and non-obese subgroups. Compared with obese IL-6-low asthmatics, the obese IL-6-high asthmatic group had strong signals for metabolic dysfunction, such as history of hypertension, and increases in total blood leukocytes and blood neutrophils. Moreover, asthma outcomes were significantly worse in obese IL-6-high asthma than in obese IL-6-low asthma. In addition, indicators of metabolic dysfunction and more severe asthma were also characteristics of non-obese IL-6-high asthmatics, indicating that IL-6 is associated with metabolic dysfunction and severe asthma, even in the absence of obesity [93].

Collectively all these observations emphasize the relevance of metabolic dysfunction in the association of asthma and obesity. These findings help explain why obesity is associated with severe asthma in some, but not all obese asthmatics. The data collected from various studies clearly support the inclusion of the evaluation of metabolic dysfunction parameters in any investigation addressed at examining the links between obesity and asthma.

## 6. Effects of Weight Loss on Asthma and Inflammation

Weight loss obtained thanks to dietetic treatments has a beneficial effect on asthma control, and reduces the use rescue medication, asthma exacerbations, hospitalizations, as well as improving the quality of life and lung function tests [177,178,179]. However, when assessed, there were no significant changes in markers of inflammation including: FeNO, induced sputum cellularity, leptin, C-reactive protein, eotaxin, and TGF-β [177,178]. Combining diet with exercise is more effective on asthma control than taking these measures alone [180,181,182]. Inflammatory leptin and IL-6 plasma levels were significantly reduced after the combined interventions [180]. Percentage sputum eosinophil was significantly reduced in subjects who completed the diet-exercise intervention. In contrast, a significant reduction in sputum percentage neutrophils was found in females but not in males [180]. A significant reduction in plasma levels of IL-4, IL-6, TNF-α, and leptin, associated with increased levels of 25-hydroxy vitamin D (25(OH)D), IL-10, and adiponectin has also been reported with the combined therapy [181,182].

Weight loss between 5% and 10% seems sufficient to significantly improve asthma control both with diet alone and combining dietetic and exercise therapy [177,178,179,180,182].

Bariatric surgery is a surgical technique that involves reducing the volume of the stomach or decreasing the absorption of food by excluding part of the intestine. There are four main surgical procedures that are very effective in causing significant weight loss: The adjustable gastric band, gastric bypass, sleeve gastrectomy, and biliopancreatic deviation. The beneficial effect of bariatric surgery on severity, control, or therapeutic load was reported in persistent moderate/severe asthma in the late 1990s [183,184]. These seminal studies found significant improvements in asthma control, asthma severity, medications required, hospitalizations, sleep, exercise capacity, and lung function tests. The significant improvement in asthma control persists five years after surgery [183,184].

In a large retrospective study, bariatric surgery, whatever the procedure, led to the discontinuation of all drugs in 39.3% of patients within one year. Bronchodilators were no longer used by 42% of the patients and 41% could also discontinue inhaled corticosteroid therapy [185]. Similar results, with a 49% reduction in inhaled treatments at one year of bariatric surgery, improvement in asthma control assessed by the asthma control questionnaire (ACQ) score, and reduction of emergency room visits were found in other studies [186,187]. Bariatric surgery also improved lung function and reduced AHR [188,189]. However, in another study, the improved effect on AHR was only found in the group of obese asthmatics with high serum (IgE) levels [187]. The effect on asthma control, therapeutic reduction, and lung function seems to persist after five years of bariatric surgery [190].

With respect to markers of allergic inflammation, no changes in submucosal cell counts of eosinophils, neutrophils, B cells, macrophages, CD4+ T cells, or CD8+ T cells were found with weight loss bariatric surgery. In contrasts, mast cells decreased significantly in the same patients [160].

The mechanisms that potentially can link obese- and- asthma-related inflammation remain to be elucidated. A recent study sought to determine if AT inflammation was associated with increased inflammation in the airway of obese asthma patients and whether weight loss after bariatric surgery would simultaneously improve metabolic and airway inflammation. Visceral AT from obese subjects with asthma isolated at the time of bariatric surgery had significantly lower adiponectin, but higher leptin and CD68 (a macrophage marker) expression compared with control participants, and this was independent of BMI. AHR was more significantly related to visceral fat leptin than to BMI. In the obese subjects with asthma, concentrations of the pro-inflammatory cytokines IL-6 and IL-8 decreased with weight loss. Changes in leptin and adiponectin protein levels in airways were similar to those found in adipose tissue: At baseline leptin levels were higher, and adiponectin lower in BALF of participants with asthma compared with control subjects. The authors concluded that the high levels of adipokines produced in visceral AT in obese asthma are associated with airway reactivity but not with airway inflammation [164].

Although asthma is associated with an increased length of hospital stay [191], the use of bariatric surgery in obese asthmatics does not seem to have an impact on intervention-related mortality [192]. Taken together these observations support the use of bariatric surgery in the treatment of severe obese asthma. However, it should be considered after previous failure of medical, nutritional, dietetic, exercise, and psychotherapeutic treatment in obese patients with severe asthma.

## 7. Conclusions

It seems to be quite well demonstrated that obesity is associated with asthma and in particular with severe asthma. The mechanisms that link both processes are surely very complex. The inflammatory process underlying both diseases could be one of the potential connecting links between both diseases. However, the true relevance that inflammation can exert as a bridge between both processes remains to be elucidated. Because asthma is a heterogeneous disease, and obesity can be associated with metabolic disorders that may also influence asthma on their own, future research should be carried out taking into account this complexity and exclude the use of studies that want to examine the mechanisms, which link asthma and obesity using simplistic approaches. Moreover, future studies should also assess the role of the characteristics of the diet as well as exposome factors such as pollutants that can modulate the interaction between obesity and asthma.

## Figures and Tables

**Figure 1 jcm-10-00169-f001:**
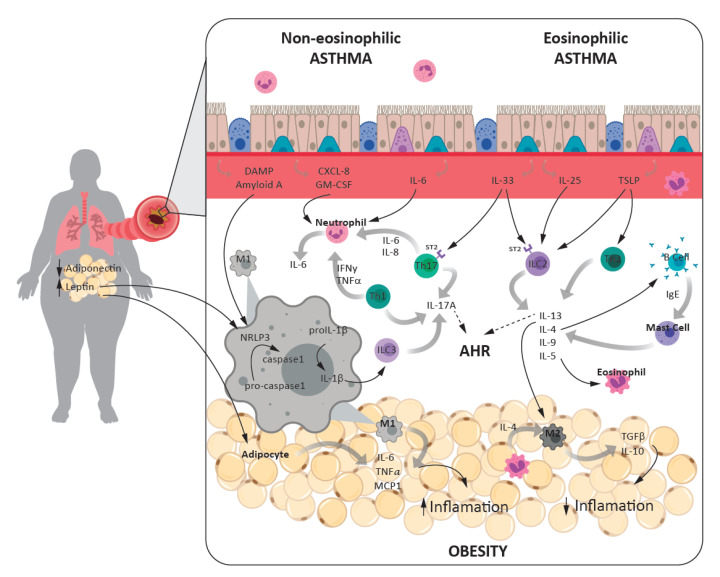
Interrelationship between the inflammatory processes that account for the asthma-obesity phenotype. Eosinophilic asthma is triggered by type 2 inflammation where Th2, ILC2, and type 2 cytokines are involved. This could be allergic (when immunoglobulin E (IgE) is present) or non-allergic. Non-eosinophilic asthma (neutrophils are present in the airway lumen) is mediated by type 1 inflammation and Th1, Th17, and ILC3 release type 1 cytokines. Inflammation in obesity is due to the imbalance of adipokines. Leptin stimulates adipocytes to release inflammatory mediators and activates M1 macrophages intracellular multiprotein complex, NLRP3. The activation of NLRP3 in M1 macrophages resident on adipose tissue and in the lungs, resulting in an amplification in IL-1β production, the subsequent ILC3 activation, and IL-17 secretion, which in turn facilitates airway hyperresponsiveness (AHR) in patients. In obesity, pro-inflammatory cytokines such IL-1β, IL-6, and TNFα contribute to asthma pathogenesis. In contrast, cytokines that are pro-inflammatory in asthma such as IL-4, IL-13, and IL-33 contribute to maintain the lean state through the activation of anti-inflammatory M2 macrophages.

## Data Availability

Not applicable.

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
