# Peer review of "Asthma and Obesity: Two Diseases on the Rise and Bridged by Inflammation"

_jcm, 2021, doi:10.3390/jcm10020169_

Round 1
Reviewer 1 Report
This is an interesting and updated review on the complex immune and inflammatory links between obesity and asthma. The manuscript is well developed and organized, although some minor changes in some of the subheading contents and a few updates on specific topics will improve the review value.
Specific points:
Introduction, lines 39-40 - The authors properly refer that BMI “is not the best marker of obesity” (line 39), but they should also recognize that the association between obesity and asthma also depends on their specific definitions (see de Castro Mendes F, et al. Asthma and body mass definitions affect estimates of association: evidence from a community-based cross-sectional survey. ERJ Open Research. 2019;5(4):00076-2019.). Additionally, what are limitations expected using BMI instead of DEXA on the asthma/obesity association?
Lines 58-61 - The authors briefly describe how diet habits might be related with asthma. It would be relevant to further explore the literature regarding the association between diet, asthma, obesity and obesity-asthma phenotype (e.g., Garcia-Larsen V, et al. Asthma and dietary intake: an overview of systematic reviews. Allergy. 2016;71(4):433-42) and some of its proposed mechanisms (e.g., Cunha P, et al. Dietary diversity and childhood asthma - Dietary acid load, an additional nutritional variable to consider. Allergy. 2020 Sep;75(9):2418-2420).
Lines 61-63 – the authors generally address the increasing prevalence of asthma in relation with environmental changes. However, and as recognized, both obesity and asthma can interact by complex and multiple mechanisms. Nevertheless, the role of environmental effects in both asthma and obesity have been less explored in the text.
It would be interesting to explore the role of the environment, namely the exposure to indoor air and green spaces, in the development of both asthma and obesity (e.g. see Dadvand P, et al. Risks and benefits of green spaces for children: a cross-sectional study of associations with sedentary behavior, obesity, asthma, and allergy. Environ Health Perspect. 2014 Dec;122(12):1329-35; Limaye S, et al. Obesity and asthma: the role of environmental pollutants. Immunol Allergy Clin North Am. 2014 Nov; 34(4): 839-55). Additionally, exposure to some indoor and outdoor chemical compounds/pollutants has also been recently associated with the development of obese asthma (e.g. Paciência I, et al. Exposure to indoor endocrine-disrupting chemicals and childhood asthma and obesity. Allergy. 2019 Jul;74(7):1277-1291; Permaul P, et al. Obesity may enhance the adverse effects of NO2 exposure in urban schools on asthma symptoms in children. J Allergy Clin Immunol. 2020 Oct;146(4):813-820.e2.)
Lines 147-149 – “IL-13 induces the proliferation of IgE-producing natural killer (NK) cells,..” – please check references 48/50 that do not seem to support this statement. In fact, IgE acting through FcγRIII, can activate NK cells resulting in cytokine/chemokine production contributing to IgE-mediated allergic responses, not to IgE synthesis (see Karimi K, Forsythe P. Natural killer cells in asthma. Front Immunol. 2013 Jun 21;4:159.)
Lines 428-443 – this subsection (3.3.4) seems somehow misplaced within the general 3.3. section. I would suggest to transfer this information (genetic links in the obesity-asthma association, pointing to inflammatory/ immune pathways) to the beginning of the introduction of section 3.3. (i.e. after line 345, deleting the subheading 3.3.4 title).
Lines 478-492 – “Effects of weight loss on asthma …” although the authors refer to some relevant Randomized Controlled Trials in this context (ref 173,174), the evidence gathered by a key systematic review on this specific issue should also be cited (Moreira A, et al. Weight loss interventions in asthma: EAACI evidence-based clinical practice guideline (part I). Allergy. 2013 Apr;68(4):425-39.).
Lines 513-525 – although described within the section of bariatric surgery, the text main message concentrates around the mechanistic pathways related to visceral AT adipokines. It may seem more adequate joining the 3.3.3 subsection (adipokines).
Fig. 1 legend, Lines 551-2 – “In contrast, cytokines that are pro-inflammatory in asthma such as IL-4, IL-13, and IL-33 contribute to maintain the lean state.” I suggest to add “…, through the activation of regulatory M2 macrophages” or “…anti-inflammatory M2 macrophages” , as represented on the bottom/ right of the figure.
Minor – please note letter type and size of subheadings titles is not uniform.
Author Response
Thank you very much for your comments and suggestions. We have taken them into account and have included them in the revised version of the manuscript. Your suggestions have been of great help to improve both the content and the presentation of the manuscript.
Point-by- point response
Reviewer (R). Introduction, lines 39-40 - The authors properly refer that BMI “is not the best marker of obesity” (line 39), but they should also recognize that the association between obesity and asthma also depends on their specific definitions (see de Castro Mendes F, et al. Asthma and body mass definitions affect estimates of association: evidence from a community-based cross-sectional survey. ERJ Open Research. 2019;5(4):00076-2019.). Additionally, what are limitations expected using BMI instead of DEXA on the asthma/obesity association?
Auhors (A). The point you make is really important. It is clear that the lack of common and standardized criteria to characterize obesity is largely responsible for many discrepant or irreproducible results. The article you recommend is about this topic and we thought it was important to include it in the manuscript.
Regarding the question about the use of DEXA or BMI in the evaluation, we do not have data to be able to give an opinion supported by scientific evidence, but we allow ourselves to speculate that any method (such as DEXA) that offers greater precision in the evaluation of excess weight will contribute to improve the scientific quality of the studies. Its implementation may be feasible in studies with small populations, however, we imagine that it can be much more complicated to be used it in epidemiological studies with hundreds / thousands of individuals
- Lines 58-61 - The authors briefly describe how diet habits might be related with asthma. It would be relevant to further explore the literature regarding the association between diet, asthma, obesity and obesity-asthma phenotype (e.g., Garcia-Larsen V, et al. Asthma and dietary intake: an overview of systematic reviews. Allergy. 2016;71(4):433-42) and some of its proposed mechanisms (e.g., Cunha P, et al. Dietary diversity and childhood asthma - Dietary acid load, an additional nutritional variable to consider. Allergy. 2020 Sep;75(9):2418-2420).
- Thank you very much for this suggestion. We were not aware of this information. The observation that in addition to the calories in diets, other aspects of them, such as their quality, can impact the development of diseases is really interesting and we have included it in the revised manuscript.
- R. Lines 61-63 – the authors generally address the increasing prevalence of asthma in relation with environmental changes. However, and as recognized, both obesity and asthma can interact by complex and multiple mechanisms. Nevertheless, the role of environmental effects in both asthma and obesity have been less explored in the text. It would be interesting to explore the role of the environment, namely the exposure to indoor air and green spaces, in the development of both asthma and obesity (e.g. see Dadvand P, et al. Risks and benefits of green spaces for children: a cross-sectional study of associations with sedentary behavior, obesity, asthma, and allergy. Environ Health Perspect. 2014 Dec;122(12):1329-35; Limaye S, et al. Obesity and asthma: the role of environmental pollutants. Immunol Allergy Clin North Am. 2014 Nov; 34(4): 839-55). Additionally, exposure to some indoor and outdoor chemical compounds/pollutants has also been recently associated with the development of obese asthma (e.g. Paciência I, et al. Exposure to indoor endocrine-disrupting chemicals and childhood asthma and obesity. Allergy. 2019 Jul;74(7):1277-1291; Permaul P, et al. Obesity may enhance the adverse effects of NO2 exposure in urban schools on asthma symptoms in children. J Allergy Clin Immunol. 2020 Oct;146(4):813-820.e2.)
- Thank you very much for providing us with this information, which once again demonstrates how complex the relationship between overweight / obesity and asthma can be. It seems clear that in future studies aimed at studying the interaction between obesity and asthma, the role of modulating factors such as pollution should be included in the equation. We have included these topics in the revised version. Given the relevance that they may have, we have also included a comment in the conclusions highlighting the need to take them into account in future studies aimed at studying the interactions between obesity and asthma.
Lines 147-149 – “IL-13 induces the proliferation of IgE-producing natural killer (NK) cells,..” – please check references 48/50 that do not seem to support this statement. In fact, IgE acting through FcγRIII, can activate NK cells resulting in cytokine/chemokine production contributing to IgE-mediated allergic responses, not to IgE synthesis (see Karimi K, Forsythe P. Natural killer cells in asthma. Front Immunol. 2013 Jun 21;4:159.)
- Thanks for warning us of this error. There has been a misinterpretation between IL-13 and IL-33. It is IL-33, not Il-13 that activates NK cells. The error has been amended.
A.Lines 428-443 – this subsection (3.3.4) seems somehow misplaced within the general 3.3. section. I would suggest to transfer this information (genetic links in the obesity-asthma association, pointing to inflammatory/ immune pathways) to the beginning of the introduction of section 3.3. (i.e. after line 345, deleting the subheading 3.3.4 title).
- Thanks for the suggestion. We agree that the location of this subsection was poorly coordinated with the previous topics. We have found a better solution to open a section for the topic (section 4) than to move it to another subsection.
Lines 478-492 – “Effects of weight loss on asthma …” although the authors refer to some relevant Randomized Controlled Trials in this context (ref 173,174), the evidence gathered by a key systematic review on this specific issue should also be cited (Moreira A, et al. Weight loss interventions in asthma: EAACI evidence-based clinical practice guideline (part I). Allergy. 2013 Apr;68(4):425-39.).
- Thanks for the suggestion. Given that the article is an excellent and relatively recent evaluation of the articles dedicated to the topic of obesity and asthma, we have found it interesting to include it in the list of bibliographic references
Lines 513-525 – although described within the section of bariatric surgery, the text main message concentrates around the mechanistic pathways related to visceral AT adipokines. It may seem more adequate joining the 3.3.3 subsection (adipokines).
- Thanks for the suggestion, but in this case we respectfully disagree with your opinion. The way it is presented seems more logical to us than the one you propose.
Fig. 1 legend, Lines 551-2 – “In contrast, cytokines that are pro-inflammatory in asthma such as IL-4, IL-13, and IL-33 contribute to maintain the lean state.” I suggest to add “…, through the activation of regulatory M2 macrophages” or “…anti-inflammatory M2 macrophages” , as represented on the bottom/ right of the figure.
- Thanks for your suggestion. We have included the comment about macrophages, which helps to better interpret the figure.
Minor – please note letter type and size of subheadings titles is not uniform.
- Thanks for warning us about these mistakes. We have unified sizes and subheadings.

Reviewer 2 Report
In this article, “Asthma and Obesity: two Diseases on the Rise and 2 Bridged by Inflammation” authors reviewed a link between asthma and obesity and described its underlying mechanisms referring to recent epidemiologic research. I think that the topic is interesting and worthy to be addressed, and the manuscript is very informative, but there are some things to be edited or clarified in the manuscript as below. In addition, as you discussed extensive studies, some contents are too narrative and repetitive to understand them well. So I recommend that the manuscript should be more organized and revised those parts.
There are some issues which I want to mention.
1) 2.3 asthma and obesity: two linked diseases Line 97-103
Some could argue that just asthma medication, especially oral corticosteroid, not asthma by itself may contribute to obesity in the other reference (Sweeney J, et al. Thorax 2016;71:339–346). So it would be better that you add related contents about the medication for asthma from your reference.
2) 3.1. Asthma and inflammation, Line 129-131
You mentioned that the majority of studies on the obese asthma phenotype use the eosinophilic, non-eosinophilic terminology. As phenotyping of asthma to eosinophilic or non-eosinophilic is applied to all types of asthma, not just limited to obese asthma, so I suggest that you add references in the sentence
3) 3.1.1 Eosinophilic asthma Line 137-138
I think that the reference No 45 is not proper to refer to the importance of both innate and adaptive immunity in the immunologic mechanisms in asthma, because at a glance it is a study of innate and adaptive immunity in the immunologic mechanisms in inflammatory bowel disease.
4) 3.1.2 Non-eosinophilic asthma, Line 194-195
I think that you insert some words to make the sentence more adequately, like this. “Gene expression of NLRP3, IL-1β and caspase-1 are detected at high levels in sputum and peripheral blood of asthma patients with neutrophilic airway inflammation.”
5) 3.4 Obesity and inflammation, Line 221-222
I suggest that the meaning of the word, “metainflammation” is needed to be explained more in the sentence.
6) 3.2.2. Innate and adaptive immune systems, Line 263-264
You and other studies reported that “eosinophils and ILC2 cells causing inflammation in asthma maintain the tolerant adipose microenvironment”. Then how can it be possible that asthma may lead to obesity as you said previously. The study (Reference 32) is about childhood asthma, which is known as a typical type of eosinophilic, T2 inflammation.
7) 3.2.3 Adipokines, Line 289-291
Please check the part in the sentence whether it is correct :“Circulating serum levels of adiponection~~ and are associated with an increased release of pro-inflammatory cytokines like IL-6 and TNF-α.
8) 3.2.3 Adipokines, Line 306-308
You mention that MMW and HMW isoform in adiponectin are predominant earlier and the function of adiponectin is mainly anti-inflammatory in obesity. So it is hard to understand that due to LMW isoform, which is at very low concentration in human plasma, adiponectin has anti-inflammatory role in asthma and obesity.
9) 3.3.1 Cells and cytokines, Line 372-373
If “eosinophilic” means elevated infiltration of eosinophils in submucosa in the part of “an eosinophilic non-T2 dominant type of asthma”, you need to check if the word can be used possibly in that condition, not elevated cells in sputum or blood. I think that it could make reader confused.
10) 3.3.11 Cells and cytokines, Line 389-390
I suggest “ in asthma-> in adult asthma” because the contents in this paragraph would be partly opposite to them in the previous paragraph.
11) 5. Effects of Weight Loss on Asthma and Inflammation, Line 494~
The context is about the beneficial effect of bariatric surgery in asthma. However, in clinical practice, the operation is conducted in a subgroup with specific obesity, not anyone with obesity. So I recommend that the indication of bariatric surgery in the referring studies should be added.
12) 5. Effects of Weight Loss on Asthma and Inflammation, Line 526~530
I don’t agree with the sentence “the use of bariatric surgery in obese asthmatics does not seem to have an impact on intervention-related mortality”. As I know, bariatric surgery by itself has a morbidity and mortality, furthermore bariatric surgery in obese asthma patients has more risk of morbidity and mortality. Therefore, you should delete the sentence in 526-527.
Below are minor comments to be corrected.
1) You don’t need to insert abbreviated form (HRB) because it was used just in one time
2) There are some bold words or sentences in the whole manuscript. Please check and correct them.
3) Line 153
Nasal polys-> nasal polyps
4) Line 171~173
You don’t need insert the same reference in the consecutive sentences repeatedly (ref 65).
5) 3.2.3 Adipokines, Line 321
Please delete “transforming necrosis” in front of TNF-α.
6) 3.3.4 Association of obesity and asthma: genetics, Line 437
Please check the “effects of BMI on late-onset, atopic, and non-atopic asthma” whether it is correct.
7) 3.2.2 Innate and adaptive immune systems, Line 243
Please check the sentence which is grammarly correct.
8) 3.2.2 Innate and adaptive immune systems, Line 259
Please check the abbreviated form.
Author Response
Thank you very much for your comments and suggestions on various aspects of the manuscript. We have taken them into account in the revised version of the manuscript, which has been of great help to improve its quality.
Point-by-point response
Revisor (R). 1) 2.3 asthma and obesity: two linked diseases Line 97-103. Some could argue that just asthma medication, especially oral corticosteroid, not asthma by itself may contribute to obesity in the other reference (Sweeney J, et al. Thorax 2016;71:339–346). So it would be better that you add related contents about the medication for asthma from your reference.
Authors (A). Thank your for your comment. The frequent or regular use of oral corticosteroids is common in severe persistent asthma, which is why in these patients it can contribute to the development of obesity. In the overall group of the asthmatic population, the percentage of patients who need this treatment is very small, so it most likely does not play a relevant role in the association between obesity and asthma. For this reason, and following your suggestion, we have included in the section dedicated to obesity and severe asthma a comment highlighting the potential role that treatment with systemic corticosteroids may have in the presence of obesity associated with severe asthma.
- 2) 3.1. Asthma and inflammation, Line 129-131. You mentioned that the majority of studies on the obese asthma phenotype use the eosinophilic, non-eosinophilic terminology. As phenotyping of asthma to eosinophilic or non-eosinophilic is applied to all types of asthma, not just limited to obese asthma, so I suggest that you add references in the sentence.
- A. We logically agree with the comment that the classification has been used in asthma regardless of whether or not it is associated with obesity and that in the list of referenced articles there are several that demonstrate it and have been used in the revised version for this purpose .
- 3) 3.1.1 Eosinophilic asthma Line 137-138. think that the reference No 45 is not proper to refer to the importance of both innate and adaptive immunity in the immunologic mechanisms in asthma, because at a glance it is a study of innate and adaptive immunity in the immunologic mechanisms in inflammatory bowel disease.
- Thanks for your observation. We agree that the cited article is not appropriate. We have not been able to find an explanation for the fact that he sneaked in without us noticing. Surely, during the successive modifications that we have been making, contribute to the mistake. We have included an article appropriate to the revised text.
- 4) 3.1.2 Non-eosinophilic asthma, Line 194-195. I think that you insert some words to make the sentence more adequately, like this. “Gene expression of NLRP3, IL-1β and caspase-1 are detected at high levels in sputum and peripheral blood of asthma patients with neutrophilic airway inflammation.”
- Thank you for your comment. We have modified the text following your recommendation
- 5) 3.4 Obesity and inflammation, Line 221-222. I suggest that the meaning of the word, “metainflammation” is needed to be explained more in the sentence.
- Based on your suggestion we have included a more detailed explanation of what metainflammation means.
- 6) 3.2.2. Innate and adaptive immune systems, Line 263-264. You and other studies reported that “eosinophils and ILC2 cells causing inflammation in asthma maintain the tolerant adipose microenvironment”. Then how can it be possible that asthma may lead to obesity as you said previously. The study (Reference 32) is about childhood asthma, which is known as a typical type of eosinophilic, T2 inflammation.
- We have no answer to the dilemma you are posing us and we believe that no one does. The potential regulatory role of eosinophils in adipose tissue inflammation and its relationship to eosinophilic asthma is unclear and controversial.
- 7) 3.2.3 Adipokines, Line 289-291. Please check the part in the sentence whether it is correct :“Circulating serum levels of adiponection~~ and are associated with an increased release of pro-inflammatory cytokines like IL-6 and TNF-α.
- A. We have checked it and it seems correct
- 8) 3.2.3 Adipokines, Line 306-308. You mention that MMW and HMW isoform in adiponectin are predominant earlier and the function of adiponectin is mainly anti-inflammatory in obesity. So it is hard to understand that due to LMW isoform, which is at very low concentration in human plasma, adiponectin has anti-inflammatory role in asthma and obesity.
- We are sorry but we could not understand your comment well. It is difficult, if not impossible, to discuss LMW's role in asthma. As far as we know, his role in either asthma or obesity-associated asthma has not been investigated.
- 9) 3.3.1 Cells and cytokines, Line 372-373. If “eosinophilic” means elevated infiltration of eosinophils in submucosa in the part of “an eosinophilic non-T2 dominant type of asthma”, you need to check if the word can be used possibly in that condition, not elevated cells in sputum or blood. I think that it could make reader confused.
- Thanks for your comment. We agree that the distinction between mucosal inflammation and the presence / absence of eosinophils in sputum can be misleading. To avoid this, in the revised manuscript we point out (using parentheses) when we speak of inflammation of the submucosa to differentiate it from eosinophilic inflammation detected in sputum.
R.10) 3.3.11 Cells and cytokines, Line 389-390. I suggest “ in asthma-> in adult asthma” because the contents in this paragraph would be partly opposite to them in the previous paragraph.
- We have included the suggested change.
- 11) 5. Effects of Weight Loss on Asthma and Inflammation, Line 494~. The context is about the beneficial effect of bariatric surgery in asthma. However, in clinical practice, the operation is conducted in a subgroup with specific obesity, not anyone with obesity. So I recommend that the indication of bariatric surgery in the referring studies should be added.
- Thank you for your comment. We have specified that the indication for bariatric surgery in an obese asthmatic should be considered when asthma is persistent moderate / severe. Obviously, surgery may be indicated to treat obesity regardless of whether the patient is asthmatic or not.
- 12) 5. Effects of Weight Loss on Asthma and Inflammation, Line 526~530. I don’t agree with the sentence “the use of bariatric surgery in obese asthmatics does not seem to have an impact on intervention-related mortality”. As I know, bariatric surgery by itself has a morbidity and mortality, furthermore bariatric surgery in obese asthma patients has more risk of morbidity and mortality. Therefore, you should delete the sentence in 526-527.
- We respectfully disagree with your suggestion. There is a published article (number 192 in the reference list) that shows that bariatric surgery in obese asthmatic patients is not associated with greater morbidity and mortality than that observed in non-asthmatic obese patients.
- Below are minor comments to be corrected. 1) You don’t need to insert abbreviated form (HRB) because it was used just in one time. 2) There are some bold words or sentences in the whole manuscript. Please check and correct them. 3) Line 153 Nasal polys-> nasal polyps. 4) Line 171~173 You don’t need insert the same reference in the consecutive sentences repeatedly (ref 65). 5) 3.2.3 Adipokines, Line 321. Please delete “transforming necrosis” in front of TNF-α.. 6) 3.3.4 Association of obesity and asthma: genetics, Line 437. Please check the “effects of BMI on late-onset, atopic, and non-atopic asthma” whether it is correct. 7) 3.2.2 Innate and adaptive immune systems, Line 243 Please check the sentence which is grammarly correct. 8) 3.2.2 Innate and adaptive immune systems, Line 259 Please check the abbreviated form.
- Thank you. We have checked and corrected the errors indicated.
